# Does the Zinc Finger Antiviral Protein (ZAP) Shape the Evolution of Herpesvirus Genomes?

**DOI:** 10.3390/v13091857

**Published:** 2021-09-17

**Authors:** Yao-Tang Lin, Long-Fung Chau, Hannah Coutts, Matin Mahmoudi, Vayalena Drampa, Chen-Hsuin Lee, Alex Brown, David J. Hughes, Finn Grey

**Affiliations:** 1Division of Infection and Immunity, The Roslin Institute, University of Edinburgh, Easter Bush, Edinburgh EH25 9RG, UK; Oliver.Lin@roslin.ed.ac.uk (Y.-T.L.); L.F.F.Chau@sms.ed.ac.uk (L.-F.C.); H.Coutts@sms.ed.ac.uk (H.C.); M.Mahmoudi@roslin.ed.ac.uk (M.M.); vayalena.drampa@roslin.ed.ac.uk (V.D.); Abraham.Lee@roslin.ed.ac.uk (C.-H.L.); Alexander.Brown@roslin.ed.ac.uk (A.B.); 2Biomedical Sciences Research Complex, School of Biology, University of St Andrews, St Andrews KY16 9ST, UK; djh25@st-andrews.ac.uk

**Keywords:** herpesvirus, zinc finger antiviral protein, dinucleotide, CpG

## Abstract

An evolutionary arms race occurs between viruses and hosts. Hosts have developed an array of antiviral mechanisms aimed at inhibiting replication and spread of viruses, reducing their fitness, and ultimately minimising pathogenic effects. In turn, viruses have evolved sophisticated counter-measures that mediate evasion of host defence mechanisms. A key aspect of host defences is the ability to differentiate between self and non-self. Previous studies have demonstrated significant suppression of CpG and UpA dinucleotide frequencies in the coding regions of RNA and small DNA viruses. Artificially increasing these dinucleotide frequencies results in a substantial attenuation of virus replication, suggesting dinucleotide bias could facilitate recognition of non-self RNA. The interferon-inducible gene, zinc finger antiviral protein (ZAP) is the host factor responsible for sensing CpG dinucleotides in viral RNA and restricting RNA viruses through direct binding and degradation of the target RNA. Herpesviruses are large DNA viruses that comprise three subfamilies, alpha, beta and gamma, which display divergent CpG dinucleotide patterns within their genomes. ZAP has recently been shown to act as a host restriction factor against human cytomegalovirus (HCMV), a beta-herpesvirus, which in turn evades ZAP detection by suppressing CpG levels in the major immediate-early transcript IE1, one of the first genes expressed by the virus. While suppression of CpG dinucleotides allows evasion of ZAP targeting, synonymous changes in nucleotide composition that cause genome biases, such as low GC content, can cause inefficient gene expression, especially in unspliced transcripts. To maintain compact genomes, the majority of herpesvirus transcripts are unspliced. Here we discuss how the conflicting pressures of ZAP evasion, the need to maintain compact genomes through the use of unspliced transcripts and maintaining efficient gene expression may have shaped the evolution of herpesvirus genomes, leading to characteristic CpG dinucleotide patterns.

## 1. Introduction

Genomes display large variability in their nucleotide composition, including in the frequencies and quantity of certain nucleotide pairs such as CpG and TpA (termed dinucleotides where consecutive nucleotides are linked by a phosphate bond on the same DNA strand) [1]. Compositional genomic biases occur due to redundancy within the genetic code. Each amino acid has between two and six alternative codons, except methionine and tryptophan which are encoded by a single codon. Due to this redundancy, changes in the third nucleotide position of a codon often result in synonymous mutations, in which a change to the nucleotide sequence does not lead to a corresponding change in the amino acid sequence (which differs from non-synonymous mutations that alter the amino acid sequence). Accumulation of synonymous mutations can lead to genome compositional bias that can have fundamental effects on gene expression while leaving protein-coding unchanged. These include nucleotide, dinucleotide, codon usage and codon pair bias.

As viruses are obligate intracellular pathogens, they are dependent on the cellular translational machinery and in some cases, such as herpesviruses, the transcriptional machinery. They are therefore subject to the same effects caused by genomic compositional bias. Nevertheless, such biases were shown to play an important role in cellular defences against viruses, while viruses also manipulate compositional biases within their own genomes to evade host defence mechanisms [2].

Herpesviruses are double-stranded DNA viruses with relatively large genomes that are highly disseminated in nature and can be found in almost all vertebrates and some invertebrates. There are currently over 150 viruses described, including 8 clinically important human herpesviruses and several significant veterinary pathogens [3].

Millions of years of co-evolution have shaped Herpesviruses into masters of immune evasion [4,5]. A central characteristic of herpesviruses is their ability to maintain life-long infections despite robust antiviral responses from the host. To achieve this, herpesviruses express multiple accessory proteins, many of which are dedicated to the evasion of host restriction factors [6]. These viral evasion genes display the characteristic evolutionary signatures of ongoing conflict with host defence [7,8].

In this review, we will discuss the role of compositional genomic bias in herpesvirus genomes and how the targeting of viral RNA with high CpG dinucleotide frequencies by ZAP may have shaped the evolution of herpesvirus genomes.

## 2. Nucleotide and Dinucleotide Bias

Nucleotide content can vary drastically between different species and within a genome. For example, the GC content in human genomes can vary between approximately 35% to 60% over a 100 kb region, with high GC content associated with transcriptionally active regions, whereas AT-rich regions tend to be associated with heterochromatic regions with low gene density [9]. Nucleotide biases also occur in viruses, with herpesviruses displaying large disparities in GC content, ranging from 37% for human herpesvirus 7, compared to 70% for Herpes simplex virus 2 (Figure 1A) [10,11]. In contrast to mammalian genomes, the vast majority of herpesvirus genomes are transcribed, meaning genomic nucleotide content has a direct consequence for transcript nucleotide content with a clear corresponding relationship between the two (Figure 1B).

In addition to biases in overall nucleotide content, dinucleotide bias occurs when pairs of nucleotides occur more or less frequently than would be expected based on the overall nucleotide content of a gene or genome. Among these, CpG dinucleotides have been most extensively studied. CpG dinucleotide frequencies refer to the expected number of CpG sequences within a gene or genome compared to the overall GC content. For example, plant and vertebrate genomes show significantly lower CpG frequencies than would be expected given their overall base composition [12,13,14]. This is thought to arise through spontaneous deamination of methylated cytosines in nuclear DNA, causing conversion to thymine and resulting in CpG sequences mutating to TpG [15].

Previous studies have demonstrated significant suppression of CpG and UpA/TpA dinucleotide frequencies in the coding regions of RNA and small DNA viruses, which mimics suppression in mammalian genomes [12]. Artificially increasing dinucleotide frequencies results in a substantial attenuation of virus replication, suggesting dinucleotide bias facilitates recognition of non-self RNA and that viruses have evolved to counter this [2,16,17,18]. Recently, the antiviral protein ZAP (also known as PARP13), expressed from the ZC3HAV1 gene, was identified as the host factor responsible for sensing CpG dinucleotides in viral RNA [19].

## 3. Zinc Finger Antiviral Protein (ZAP)

The role of ZAP as a host restriction factor has recently been comprehensively reviewed [20]. Briefly, there are four isoforms of ZAP; ZAP short (ZAPS), ZAP medium (ZAPM), ZAP long (ZAPL) and ZAP extra-long (ZAPXL), generated by alternative splicing and polyadenylation [21]. The predominant forms are ZAPS and ZAPL. While ZAPS is induced by IFN signalling, ZAPL is constitutively expressed, presumably due to differential splicing downstream of IFN signalling, although how this occurs is currently unclear. ZAPL contains an enzymatically inactive PARP-like domain at the C terminal. This Domain is S-farnesylated at Cys-899, directing localisation to the endolysosomes and endoplasmic reticulum and is required for its antiviral activity [22]. Both ZAPS and ZAPL have antiviral activity, although levels of inhibition vary against different viruses [20]. ZAP antiviral activity is based on direct binding to viral RNA and has been linked to degradation through recruitment of the exosome complex components and inhibition of translation [23,24,25].

ZAP was originally identified as a host restriction factor by screening a rat cDNA library for proteins that inhibited the replication of Moloney murine leukaemia retrovirus [24]. ZAP has since been shown to have antiviral activity against a range of RNA viruses, including HIV, filoviruses, flaviviruses, coxsackievirus B3, influenza A virus, Newcastle disease virus and SARS-CoV-2 [26,27,28,29,30,31,32]. More recently, we and others have shown that ZAP can restrict the replication of large DNA viruses. Disruption of ZAP increases replication of the herpesvirus HCMV and the poxvirus vaccinia [33,34,35]. In the case of the vaccinia virus, the viral C16 protein was shown to antagonise ZAP antiviral activity by relocalising ZAP to punctate cytoplasmic structures. Without C16, the virus was significantly attenuated but could be rescued by deletion or knockdown of ZAP.

There is now considerable evidence that ZAP is responsible for targeting transcripts with high CpG content. In 2017, Takata et al. showed that knockdown of ZAP rescued the replication of HIV constructs with artificially increased CpG levels introduced through synonymous mutations [19]. CLIP-seq studies using a tagged version of ZAPS demonstrated biding to regions of HIV RNA with elevated CpG dinucleotide frequencies, indicating direct recognition of CpG dinucleotides in target RNA. ZAP Knockdown also rescued attenuated echovirus 7 and Zika with artificially increased CpG levels [17,36]. Crystal structure studies also support a role for ZAP biding to CpG dinucleotides with the four zinc fingers of ZAP forming a binding pocket that favours interaction with CpG dinucleotides [37]. So far, it is unclear which isoform of ZAP is responsible for targeting transcripts with high CpG dinucleotide frequencies. Takata et al. reported that ZAPS specifically binds to CpG dinucleotide motifs in viral RNA. Our studies showed that both ZAPS and ZAPL were able to inhibit the expression of HCMV transcripts with high CpG content [34].

Although binding to CpG motifs are clearly important to the antiviral activity of ZAP, precise biding specificities have not yet been fully elucidated. Sequence context plays an important role in ZAP binding, as increased CpG frequency by itself does not necessarily predict ZAP antiviral activity. Studies comparing natural sequence variation in different strains of human and simian immunodeficiency virus found ZAP sensitivity correlated with CpG levels within the 5′ end of the viral *env* gene, but not within the rest of the genome [38]. Furthermore, sensitivity to ZAP correlated with minimum free energy levels of the RNA, suggesting secondary structure may also play an important role. This is supported by structural studies which showed the binding pocket formed by the zinc fingers in ZAP would limit association with higher-order RNA secondary structure [37]. Furthermore, while the introduction of CpGs into the 5′ end of the *env* gene increased viral sensitivity to ZAP and IFN, the introduction of CpGs into other regions of the viral genome resulted in ZAP independent attenuation, through alteration of splicing and methylation-specific inhibition of transcription, suggesting that suppression of CpG dinucleotides in viral genomes may be due to multiple factors [39].

In addition to the impact of wider sequence context, the bases directly flanking CpG dinucleotides have also been implicated in ZAP targeting, suggesting an extended binding motif. While uracil (U) bases on either side of CpG dinucleotides increased ZAP binding and viral inhibition, the presence of Adenine (A) bases flanking CpG dinucleotides disrupted ZAP binding.

Other dinucleotides have also been implicated in ZAP antiviral activity. Studies in Echovirus 7 and Zika have shown that attenuation of virus replication through the introduction of UpA dinucleotides can be rescued by deletion of ZAP [17,36]. It is yet to be determined whether ZAP binds directly to UpA, however, these studies suggest that ZAP antiviral activity is dictated by a broader spectrum of sequence motifs than CpG frequency alone. There does not seem to be competition for binding between UpA and CpG, suggesting ZAP may bind these dinucleotides through independent mechanisms.

Multimerisation of ZAP, or associated factors, is likely to play a role, given the importance of CpG frequency, also suggesting spacing between CpG motifs may be a factor. While progress has been made in understanding the role of ZAP in targeting transcripts with characteristic nucleotide frequency, more studies are required to enable the accurate prediction of ZAP targets through sequence analysis alone.

Associated factors that regulate ZAP antiviral activity were also discovered, with TRIM25 and KHNYN identified as cellular co-actors. TRIM25 is an E3 ubiquitin ligase and an ISG with antiviral activity in its own right [40]. It directly interacts and ubiquitinates ZAP while triggering self-ubiquitination [41,42]. However, ubiquitination does not seem to be required for ZAP antiviral activity and the role of TRIM25 in ZAP function is unclear. TRIM25 may, however, be involved in regulating ZAP transcription. We recently demonstrated that TRIM25 is required for efficient ZAPS induction in response to IFN signalling, through modulating differential splicing of ZAPS and ZAPL, suggesting one way in which TRIM25 may contribute to ZAP activity [34]. In addition to TRIM25, KHNYN was identified as a cofactor for ZAP through a yeast-two-hybrid screen [43]. Depletion of KHNYN was able to rescue HIV constructs containing regions of high CpG sequence, phenocopying the effects of ZAP. ZAP is not thought to directly inhibit the expression of target RNAs, instead, it interacts with other cellular factors that degrade the target RNA or inhibit translation. Viral attenuation through the introduction of CpG and UpA dinucleotides can be rescued through knockout of the cellular endonuclease RNAse L and 2′-5′-oligoadenylate synthetase 3, suggesting these genes play an important role in ZAP antiviral activity and are potentially important for the degradation of target RNAs [36].

## 4. CpG Profiles of Herpesviruses

Recently, we identified ZAP and TRIM25 as host restriction factors against HCMV using an arrayed interferon-stimulated gene expression screen, the first such screen against a herpesvirus [34]. These results were later confirmed by an independent study that showed CRISPR Cas9 deletion of ZAP increased HCMV replication and that ZAP targeted transcripts from the UL4-UL6 region of the virus [33].

Given the recent findings on ZAP targeting, the CpG profile of herpesvirus transcripts raises interesting questions about how they have evolved to counter ZAP restriction. Herpesviruses are classified into three subfamilies: alpha, beta and gamma, based on biological properties, including host range, immune evasion mechanisms, pathogenesis, and the site of latency. This classification is further supported by nucleotide sequencing, which contributes to the further categorisation of the herpesviruses into genera, based on DNA sequence similarity and protein homologues [44]. The alpha-herpesvirus subfamily (e.g., herpes simplex virus 1, HSV-1) has general characteristics of a broad host range, a relatively short replication cycle that allows the virus to spread rapidly and cause destruction of infected cells, and the ability to establish latent infection in sensory ganglia. The beta-herpesvirus subfamily (e.g., human cytomegalovirus, HCMV) has a narrow host range, but within the host, the virus tends to disseminate throughout a variety of tissues, including secretory gland epithelium and myeloid cells. The in vitro replication cycle is also much slower than alpha-herpesviruses [44]. Gamma-herpesviruses, such as Epstein–Barr virus (EBV), have a host range that is restricted to the family or order of the natural host. In vitro, they replicate in lymphoblastoid cells, either B or T lymphocytes, and can establish latency in these cell populations. Some gamma-herpesviruses can also lytically infect particular types of epithelioid and fibroblastic cells. Latent infections of gamma-herpesviruses were found to cause lymphoproliferative diseases as some viral genes are oncogenic and can efficiently immortalise infected lymphocytes in vitro [45,46].

Strikingly, each sub-family of herpesvirus displays characteristic CpG dinucleotide frequencies within their transcripts (Figure 2) [34,47]. Most alpha-herpesviruses display uniformly high CpG levels throughout their genomes, while almost all gamma-herpesviruses have suppressed CpG levels. Beta-herpesviruses display the most striking pattern. Like alpha-herpesviruses, the majority of beta-herpesviruses transcripts have high CpG levels, except for the major immediate-early transcripts, in which CpG levels are suppressed. Caveats exist to this paradigm, as can be seen for HHV7, a beta herpesvirus that has lower CpG levels than other beta-herpesviruses and KSHV which has slightly higher CpG frequencies than other gamma-herpesviruses. However, HHV7 still displays substantial suppression of CpG dinucleotides in the major immediate-early transcript IE1, while KHSV still has relatively suppressed CpG levels compared to most alpha and beta-herpesviruses. These patterns suggest conservation through evolutionary pressure, indicating a significant functional link between CpG dinucleotide bias and the biological characteristics of each of the herpesvirus subfamilies. There was no clear pattern of bias between other dinucleotides across herpesvirus subfamilies (Figure 2B), indicating particular importance of CpG dinucleotides in herpesvirus biology. Given the recent finding that ZAP targets viral transcripts with high CpG dinucleotides, we hypothesised that the pattern of CpG dinucleotides in herpesvirus sub-families was related to evolutionary evasion of ZAP activity.

## 5. Evasion of ZAP by Alpha-Herpesviruses

The lack of significant suppression of CpG dinucleotides in alpha-herpesviruses, such as HSV-1, suggests the evolution of ZAP-specific counter-measures, as the high CpG transcripts would potentially represent strong targets for ZAP. This is supported by a previous study that showed HSV-1 was able to replicate to wild-type levels despite expression of a ZAP construct, although the construct expressed a truncated version of rat ZAP, rather than full-length human ZAP [48]. HSV-1 induces a strong host shut-off through the actions of viral proteins such as the virion host shut-off protein (vhs) [49]. As this protein is delivered with the virion as part of the tegument, this could block ZAP induction prior to virus transcription. Indeed, vhs was reported to block the expression of ZAP, although this is unlikely to be a specific mechanism targeting ZAP, rather part of the general effect of vhs on cellular gene expression [48].

## 6. Evasion of ZAP by Beta-Herpesviruses

Herpesvirus express their genes in a strict temporal manner; immediate-early transcription, followed by early and late transcription. For HCMV, this was further refined to seven temporal classes [50]. The major immediate-early genes of herpesviruses are critical trans-activators, driving acute virus replication and are thought to be pivotal in regulating viral latency [51,52]. HCMV expresses two major immediate-early transcripts, IE1 and IE2, which share the first three exons and are generated by alternative splicing and polyadenylation. As with other beta-herpesviruses, CpG levels are suppressed in the major immediate gene IE1 [34,47]. We hypothesised that HCMV, and beta-herpesviruses in general, have evolved suppressed levels of CpG in the immediate early genes as a ZAP evasion mechanism. In support of this we showed that while overexpression of ZAPS resulted in decreased expression of an early (pp52) and a late gene (pp28), both of which have high CpG levels, IE1 expression levels were not affected. Furthermore, co-expression of ZAPS and ZAPL with viral expression plasmids demonstrated limited effect on the expression of IE1, but significant reduction of pp52, consistent with suppressed CpG levels in IE1 enabling ZAP evasion. While ZAPS is induced by HCMV infection, induction only occurs in a subset of infected cells and high levels of ZAP expression was mutually exclusive with early and late gene expression. However, IE1 was expressed in cells with both high and low ZAP levels, indicating CpG suppression of IE1 allows evasion of ZAP targeting in the context of virus infection. However, demonstrating rescue of a high CpG IE1 virus is still required to confirm that evasion of ZAP is the primary reason for suppressed CpG dinucleotides in the IE1 region.

The lack of suppression of CpG dinucleotides in the HCMV genes expressed subsequent to IE1, suggests a loss of pressure to maintain low CpG levels in these genes, possibly due to the loss of ZAP targeting. HCMV effectively blocks IFN signalling early in infection, in part through the actions of IE1 and IE2, suppressing ZAPS induction. However, initial western blot analysis showed high levels of ZAP expression throughout HCMV infection of cells. This would be counterintuitive based on the CpG pattern of the virus, as ZAPS would be predicted to target early and late genes, effectively inhibiting virus replication. However, separation of cells by sorting into high and low infection levels based on viral GFP expression, showed that cells with productive infections had low levels of ZAP expression, while high levels of ZAP expression occur in abortively infected cells, indicating a population-specific pattern of ZAP expression missed by standard bulk western blot approaches. Rather than a ZAP specific phenomenon, this likely reflects a balance between the ability of the virus to successfully shut down the IFN response, versus the host cell mounting a response that is able to inhibit the progression of virus replication at an early stage.

The precise mechanism by which ZAP inhibits HCMV replication is still unclear. A recent study identified transcripts from the UL4–UL6 region as specific targets of ZAP during infection with HCMV. Analysis of total and newly synthesised viral RNA showed accelerated expression of UL4–UL6 transcripts in ZAP knockout cells, while CLIPseq analysis identified ZAP binding sites within these transcripts. Why ZAP specifically targets these transcripts is unclear, as there appears to be a limited difference in dinucleotide content in UL4–UL6 transcripts compared to other HCMV transcripts, other than IE1, which has suppressed CpG levels. UL4–UL6 is among the most abundantly transcribed regions of the HCMV genome, which may explain the preferential binding of ZAP. It is also unclear why the virus has not evolved to evade targeting in this region. One possible explanation is that these transcripts act as a decoy target for ZAP, contributing to ZAP evasion, although further studies will be required to answer these questions [33].

## 7. Evasion of ZAP by Gamma-Herpesviruses

Similar to RNA viruses and small DNA viruses, the majority of gamma-herpesviruses suppress CpG dinucleotides throughout their genomes, which would be predicted to evade targeting by ZAP. Despite this, murine gamma-herpesvirus-68 (MHV68) is susceptible to ZAP in tissue culture, with ZAP specifically targeting regions within *M2* and *ORF64* [53,54]. Both have higher CpG frequencies compared to other MHV68 transcripts, but relatively low levels compared to the majority of alpha and beta-herpesvirus genes [34]. Replication and transcription activator (*RTA*), which encodes the viral protein necessary to initiate the lytic cycle, was also reported to inhibit ZAP activity, but as yet, the mechanism involved is unclear [54].

## 8. CpG Dinucleotide Bias, Gene Expression Efficiency and Splicing

While suppression of CpG dinucleotides provides a clear fitness benefit to viruses due to evasion of ZAP targeting, questions remain as to why the majority of alpha and beta-herpesvirus genes maintain relatively high levels of CpG dinucleotides. While studies indicate that high CpG abundance alone does not predict ZAP binding, artificial increases in CpG frequencies in other viruses clearly leads to ZAP dependent attenuation. One could hypothesise that natural drift towards neutral CpG frequencies, based on overall GC content occurs due to lack of evolutionary pressure following successful evasion of ZAP. However, some alpha and beta-herpesviruses have low GC content (Figure 1 and Figure 2) suggesting simple drift is an unlikely explanation. An alternative possibility is that, despite the potential for ZAP targeting, high CpG levels provide a fitness benefit to alpha and beta herpesviruses, resulting in evolutionary pressure against CpG suppression in the majority of their genes.

This pressure may be due to the fundamental effects of nucleotide content on gene expression efficiency, especially in unspliced transcripts. As well as removing introns, splicing enhances nuclear export by facilitating the association of transcripts with the cellular RNA export machinery [55,56]. A recent study indicated that high GC content in unspliced transcripts can increase expression efficiency through enhanced export to the cytoplasm [57]. While this study did not differentiate between the two, it is plausible that effects on gene expression are specifically due to CpG frequency, rather than GC content.

Limits on viral genome size, imparted by the physical constraints of capsid packaging, means the majority of herpesvirus genes are unspliced. This creates conflicting pressures on the nucleotide content of herpesvirus transcripts, with low CpG dinucleotide frequencies enabling ZAP evasion but potentially contributing to inefficient export and expression of unspliced transcripts. These forces may have shaped the evolution of herpesvirus sub-family genomes in different ways depending on the mechanisms employed to evade ZAP while maintaining efficient viral gene expression.

However, additional factors may have also influenced the nucleotide content of herpes viruses. Epigenetic regulation of the viral genome has been implicated in the establishment, maintenance and reactivation of herpesviruses from latency [58]. The same process of methylation and spontaneous deamination that results in suppression of mammalian genomic CpG frequency has been suggested to cause the specific patterns of CpG frequencies observed in sub-families of herpesviruses. Pervasive methylation of CpG dinucleotides is observed in latent KSHV and EBV genomes [59,60]. In contrast, HSV-1 genomes from latently infected dorsal root ganglia of mice showed little evidence of extensive CpG methylation, which would allow accumulation of CpG dinucleotides due to lack of spontaneous deamination [61]. The role of DNA methylation in beta-herpesviruses is less clear as fewer studies directly addressing this issue exist. However, unlike alpha and gamma-herpesviruses, DNA methylation would have to occur specifically within the immediate early open reading frame to explain the specific pattern of CpG dinucleotides associated with beta-herpesviruses. Expression of transgenes driven by the HCMV major immediate early promoter (MIEP) in the context of adenovirus vectors delivered to the muscle tissue of rats, were silenced in a matter of days by DNA methylation, suggesting methylation-specific regulation of the MIEP of HCMV [62]. However, regulation of gene expression is usually linked to CpG methylation in the promoter region, whereas suppression of CpG dinucleotides are associated with the coding region of IE1 and not the promoter region [34]. This argues against the distinctive pattern of CpG dinucleotides in HCMV being related to epigenetic regulation of gene expression.

Direct methylation of RNA can also regulate gene expression by modulating RNA splicing, export, stability and translation [63]. While N^6^-methyladenosine (m^6^A) is the most abundant internal RNA modification of cellular mRNAs, N^5^-methylcytosine (m^5^C), was reported to enhance mRNA nuclear export through the recruitment of Aly, a nuclear mRNA export adaptor [64]. Changes in CpG dinucleotide frequency could alter post-transcriptional methylation modifications of viral RNA impacting gene expression efficiency.

The role of codon bias and codon pair bias in virus replication and gene expression was also investigated [2]. Synonymous codons of the same amino acid do not occur with equal frequency and reflect the variable concentration of tRNAs within a cell, with highly expressed genes preferentially encoding codons that correspond to abundant tRNAs. This leads to some codons being used more often than others. Codon pair bias refers to the observation that some codon pairs occur more or less frequently than would be expected in protein-coding genes and the addition of rare codon pairs can negatively impact gene expression efficiency. Recoding of RNA viruses to include higher frequencies of rare codon pairs was shown to cause attenuation, without affecting protein coding [65]. To date, studies have shown low codon bias in herpesviruses while codon pair bias has not been investigated [66,67]. However, a recent study that analysed 1520 vertebrate viruses, including herpesviruses, found that codon usage biases depend on whether the virus genome was DNA or RNA, strandedness and the specific replication compartment of the virus (nucleus versus cytoplasm), suggesting a complex interplay between codon usage bias and fundamental characteristics of individual virus species [68].

## 9. Enhancement of Unspliced Viral Transcripts by a Family of Herpesvirus RNA Binding Proteins

Gene expression is reliant on successive, interdependent post-transcriptional RNA processing events including splicing, polyadenylation, nuclear export and translation [69]. The compartmentalisation of eukaryotic cells enables strict quality control on RNA export from the nucleus to the cytoplasm through the nuclear pore complex (NPC). Multiple export pathways exist for different classes of RNAs, involving specific sets of ribonucleoprotein complexes and adaptor proteins [70]. tRNAs are directly bound by the exportin-t protein, whereas microRNA export is mediated by exportin-5. The CRM1 complex is responsible for the export of small nuclear RNAs (snRNAs) and ribosomal RNAs (rRNAs) and a subset of mRNAs that contain AU rich elements (AREs) in their 3′UTRs. Bulk mRNA export occurs primarily through the Transcript and export (TREX) complex. TREX comprises the THOC sub-complex, Aly and the RNA helicase UAP56 and enables the export of mRNA transcripts from the nucleus to the cytoplasm in a splicing dependent fashion, through binding to the cap-binding complex and splicing factors [71,72]. TREX recruits the TAP/NFX1 complex, which in turn translocate transcripts across the nuclear pore complex into the cytoplasm [73]. The lack of splicing factors associated with unspliced messenger transcripts leads to poor association with the nuclear export machinery and inefficient expression.

Based on data from HSV-1, alpha-herpesviruses counter ZAP targeting immediately after infection, while beta-herpesviruses such as HCMV evade ZAP by suppressing CpG levels in the earliest transcripts before suppressing ZAP induction. Splicing of the immediate early genes ensures efficient expression despite suppressed CpG levels. A different scenario confronts gamma-herpesviruses, where CpG suppression occurs throughout the genome. While effectively evading potential targeting by ZAP, low CpG levels in unspliced genes could lead to inefficient expression, reducing viral fitness [57]. To counteract these effects, herpesviruses encode a family of RNA binding proteins that specifically enhance the export of unspliced transcripts. The best studied of these are ICP27 of HSV-1 and ORF57 of Kaposi’s sarcoma-associated herpesvirus (KSHV).

Both ICP27 and ORF57 directly recruit components of the TREX complex to unspliced viral transcripts, bypassing the requirement for splicing dependent recruitment and ensuring efficient translocation across the nuclear pore complex and into the cytoplasm [74,75]. Of the HCMV homologues, UL69 is less well studied, however, it was also shown to interact with components of the TREX complex and is required for efficient export of unspliced viral transcripts [76].

While all three proteins share a central functional role in facilitating the export of viral RNA, it is interesting to speculate whether these related viral proteins have evolved divergent RNA binding specificities in response to the characteristic dinucleotide content of each subfamily of herpesvirus. Multiple studies have attempted to determine consensus RNA binding motifs for ICP27 and ORF57. While binding sites with specific motifs were identified in a minority of viral and cellular transcripts [77], studies suggest that binding may be based on more general aspects of nucleotide content.

Early studies with ICP27 demonstrated a propensity for binding to G homopolymer substrates compared to other RNA homopolymers [78]. Further studies using a panel of oligonucleotide binding substrates and electrophoretic mobility shift assays also supported a preference for GC rich sequences [79], while genome-wide approaches confirmed a general preference for ICP27 binding to GC rich sequences. High throughput RNA sequencing of HSV-1 infected cells and cells expressing ICP27 showed altered splicing and polyadenylation of a subset of cellular genes with high GC content, while direct identification of sequences bound to ICP27 by cross linking and immunoprecipitation (CLIP-seq) also showed a preference for binding to GC rich RNA sequences [80,81]. While these studies were investigating the role of ICP27 in altering splicing and polyadenylation, rather than the export of unspliced transcripts, they are consistent with a general binding preference for GC rich RNA sequences.

There are fewer studies directly addressing the binding specificities of ORF57. However, similar to ICP27, ORF57 binding preferences match the GC nucleotide content of the viral genome [80,81]. K15 is a KSHV gene that contains introns and has a low GC content. Co-transfection of K15 with ORF57 resulted in modest enhancement of expression. However, co-transfection of a cDNA version of K15, in which the introns had been removed, resulted in far greater enhancement of expression, consistent with the poor expression of intronless transcripts with low GC content. Critically, increasing the GC content of K15 cDNA rescued expression and resulted in ORF57 independent expression. A similar pattern was observed for other KSHV genes. Again, while these studies focused on GC content, the potential role of CpG dinucleotide frequencies was not considered and may play an important role in determining target specificity.

Expression of ICP27 and its functional homologues may therefore have evolved, in part, to enable resolution of the conflicting pressures of evading ZAP detection, while maintaining efficient expression of unspliced transcripts. However, questions remain as to how this relates to the specific patterns of CpG dinucleotides in the different sub-families of herpesviruses. If ORF57 enables the efficient expression of unspliced transcripts with low CpG levels, why do the majority of alpha and beta herpesvirus transcripts maintain high CpG frequencies, potentially subjecting them to ZAP targeting? One possibility is that by maintaining a dinucleotide bias that is different from the host, it enables preferential export of viral transcripts over host transcripts through CpG specific binding by ICP27 or UL69 in the case of HSV-1 or HCMV. If this is the case, then gamma-herpesviruses, such as KSHV, must have developed an alternative method for differentiating between host and viral transcripts, as ORF57 would be unable to differentiate between host and viral transcripts based on CpG content. Alternatively, the different patterns of CpG dinucleotides may simply reflect divergent strategies of overcoming conflicting pressures of maintaining efficient viral gene expression while avoiding ZAP detection in the ancestral viruses of alpha, beta and gamma-herpesviruses.

So far, studies aimed at understanding the binding preferences of ICP27 and ORF57 have not differentiated between GC content and CpG dinucleotide frequencies. While some alpha herpesviruses have much higher GC content compared to the host, others, such as Varicella Zoster Virus (VZV) have significantly lower GC content. Only CpG dinucleotide patterns are relatively uniform within the herpesvirus subfamilies. It would be interesting to compare the sequence binding preferences of ICP27 homologues from HSV-1 and VZV. Binding preferences of the ICP27 homologues would be different if GC content was the important factor, whereas they would be similar if based on CpG dinucleotide frequencies. Given only CpG dinucleotide frequencies are uniform across alpha-herpesviruses, and within herpesvirus sub-families in general, we would predict that binding preferences would be based on CpG frequencies, rather than GC content.

## 10. Concluding Remarks

These studies are consistent with an evolutionary paradigm in which conflicting pressures shape the nucleotide content of herpesviruses, resulting in characteristic and specific patterns of CpG frequency within each subfamily (Figure 3). Alpha-herpesviruses have uniformly high CpG frequencies throughout their genomes, consistent with a lack of evolutionary pressure from the antiviral activity of ZAP. This suggests that the virus is able to block ZAP targeting of high CpG viral transcripts at a very early stage of infection, potentially through host shut off mechanisms associated with vhs tegument protein delivered with the virion, and active before the first viral transcript is generated. In general, alpha-herpesviruses also replicate more rapidly than beta and gamma-herpesviruses, potentially enabling the virus to “outrun” host antiviral responses.

Beta-herpesviruses do not trigger host shut off in infected cells and therefore need to contend with ZAP targeting immediately after infection. Our findings suggest HCMV, and other beta-herpesviruses, achieve this by suppressing CpG levels in the major immediate early genes, among the first transcripts expressed after infection. Immediate early proteins, including IE1, facilitate a block in IFN induction, reducing ZAPS expression. A caveat to this model is the potential antiviral role of ZAPL, which is constitutively expressed and would not be affected by a block in IFN signalling. However, data from our paper show reduced levels of both ZAPS and ZAPL in cells productively infected with HCMV, suggesting inhibition of ZAP expression may be playing a role [34].

Gamma-herpesviruses have evolved low CpG frequencies throughout the genome, thereby reducing potential targets for ZAP.

To maintain compact genomes, most herpesvirus genes are unspliced. While saving genomic space the general inefficiency in the expression of unspliced transcripts would potentially reduce the overall fitness of herpesviruses. Alpha- and beta-herpesviruses may have overcome this, to some extent, by maintaining higher CpG levels in the majority of their genes, which may increase the efficiency of expression, particularly in unspliced transcripts. All three herpesvirus subfamilies express RNA binding proteins that enhance gene expression by shuttling unspliced viral transcripts from the nucleus to the cytoplasm. This may be particularly important for gamma-herpesviruses, given the low CpG content of their genes. While not conclusive, studies also support specific adaptation of this family of RNA binding proteins to the unique nucleotide content of each herpesvirus subfamily, with ORF57 preferentially binding low GC transcripts, while ICP27 binds to transcripts with high GC content.

A distinction between GC content and CpG frequencies can be difficult to disentangle. While GC content and CpG frequency are of course naturally linked, an increase in one does not always result in an increase in the other. For example, while there are clear distinctive patterns of CpG frequencies within herpesvirus subfamilies, no such pattern of GC content is apparent, suggesting a lack of clear biological relevance to GC content that would be linked to the evolution of the viruses. Further studies will be required to determine whether GC content or CpG frequencies play key roles in the expression of viral genes and binding specificities of ICP27 and its homologous.

Finally, future studies could investigate whether altering the nucleotide composition of herpes viruses through synonymous mutations could be an effective platform for the generation of rational herpesvirus vaccine candidates or vectors. Modifying codon usage bias through the introduction of synonymous mutations could allow tuneable alterations to expression levels of viral target genes and levels of attenuation, as well as increasing the visibility of viruses to the host’s immune response while maintaining the natural antigenic profile of the virus [2].

## Figures and Tables

**Figure 1 viruses-13-01857-f001:**
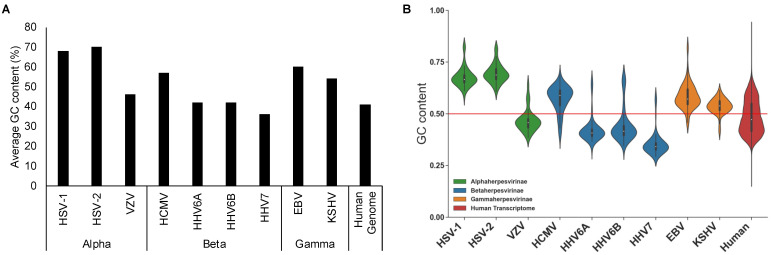
Average GC content of human herpesvirus genomes. (**A**) The average GC content was calculated based on the total number of G + C nucleotides divided by the total genome length. Calculations were based on the reference genome for each herpes virus. (**B**) GC content of human herpesvirus open reading frames and the human transcriptome. Human transcriptome data were based on GRCh38 RefSeq Transcripts. Red line indicates 50% GC content for visual reference.

**Figure 2 viruses-13-01857-f002:**
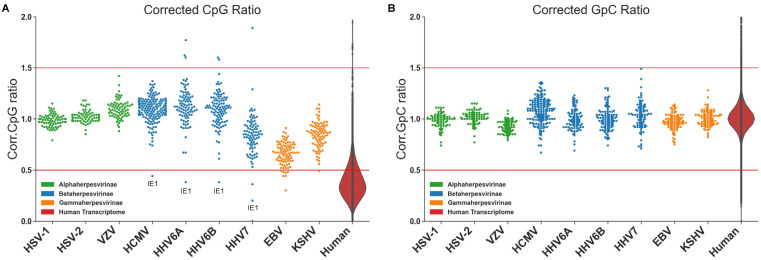
Specific suppression of CpG nucleotides within the immediate early genes of beta-herpesviruses. (**A**) The CpG content of annotated open reading frames from human herpesvirus genomes are shown, following normalisation for length and GC content. A corrected CpG ratio of one reflects the expected number of CpGs based on GC content of a transcript. (**B**) Equivalent data for GpC dinucleotide ratios are shown as a control demonstrating the CpG dinucleotide pattern is specific. No clear trend is seen with TpA (data not shown). Red lines indicate arbitrary two-fold decrease or 1.5 fold increase in dinucleotide frequency for visual comparison. Dinucleotide content of the human transcriptome is shown as a violin plot for comparison.

**Figure 3 viruses-13-01857-f003:**
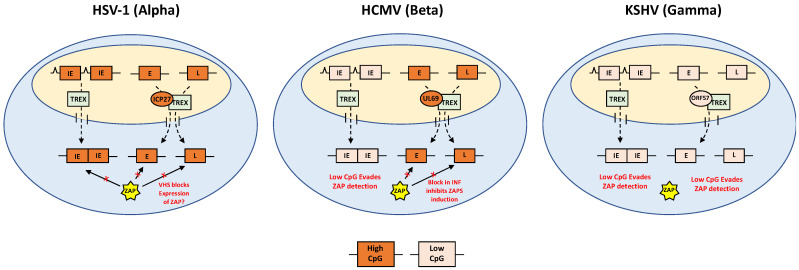
Divergent models of ZAP evasion by herpesvirus subfamilies. Uniformly high CpG frequencies throughout the HSV-1 genome suggests minimal evolutionary pressure from the antiviral activity of ZAP, possibly due to host shut-off blocking ZAP expression. Increased CpG levels would be predicted to increase expression efficiency, while ICP27 may preferentially bind to viral transcripts with high CpG frequencies. HCMV (betaherpesvirus) has suppressed CpG levels in the major immediate early gene, IE1 enabling ZAP evasion. Higher CpG frequencies in early and late genes would be predicted to increase expression efficiency while blocking IFN signalling would subvert ZAP targeting. Like ICP27, UL69 would be predicted to preferentially bind transcripts with high CpG frequencies. Low CpG frequencies throughout the KSHV genome would enable evasion of ZAP while making efficient expression of unspliced viral transcripts highly dependent on ORF57 binding. In all cases, splicing of immediate early genes would enable expression independent of ICP27 homologues.

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
