# Peer review of "Does the Zinc Finger Antiviral Protein (ZAP) Shape the Evolution of Herpesvirus Genomes?"

_viruses, 2021, doi:10.3390/v13091857_

Round 1

Reviewer 1 Report

This review by Lin et al addresses an interesting question regarding how ZAP may shape the evolution of herpesvirus genomes. Since several papers have recently characterized how ZAP targets DNA viruses, this manuscript is timely and relevant. Sections 1-3, 5 and 8 are clear and informative. The major concern for the review is in sections 6 and 7 (which appear to be identical) regarding how the authors have described their data on how ZAP restricts HCMV but not adequately described a related study from the Brinkmann lab that has different conclusions (see major comment 1). A reader who is not familiar with the field may think from this review that the Brinkmann study merely confirmed the conclusions from the authors work, which would not be correct.

Major comments

  1. One way to improve this review would be to better balance the author’s paper on ZAP restriction of HCMV (Reference 34) with the paper from the Brinkmann lab (Reference 39). There are important differences in the conclusions drawn by the two groups, but sections 6 and 7 ‘Evasion of ZAP by Beta-herpesviruses’ seem to only discuss the author’s work on this topic. The different conclusions for which HCMV transcripts are targeted by ZAP need to be discussed as well as the importance of CpG dinucleotides for ZAP binding and viral RNA degradation.

  1. It is not currently generally accepted that ZAP only targets CpG dinucleotides in viral RNA; other nucleotide motifs may also be important. The authors reference the appropriate papers by the Simmonds lab (references 16 and 32) that show that ZAP can target UpA dinucleotides in positive strand RNA viruses, but do not appear to take this into account when they imply that CpG dinucleotides are the determinant for ZAP antiviral activity. They also do not discuss work on HIV-1 that indicates the position, local sequence context or RNA structure appears to determine that CpGs in env are targeted more efficiently by ZAP than CpGs in other regions of the viral genome (Kmiec et al 2020 mBio. 11:e02903-19 and Ficarelli et al 2020 J Virol. 94:e01337-19) or that in vitro binding assays have identified an extended motif for ZAP binding (Luo et al 2020 Cell Rep. 30:46-52.e4). These points also influence the discussion in sections 9 and 10, which need to consider that nucleotide motifs other than CpG may influence ZAP antiviral activity and not all CpGs may be equally targeted by ZAP.

  1. In section 4 ‘CpG profiles of herpesviruses', the authors discuss the CpG dinucleotide frequencies within the herpesvirus DNA genomes. However, for restriction by ZAP, CpG frequencies in the RNA is the relevant factor. What is known about the CpG frequencies in hepesvirus RNAs and can this section include that information?

  1. Sections 9 and 10 are speculative and there does not seem to be much direct experimental evidence that supports the ideas regarding CpG dinucleotides and HCMV mRNA splicing and nuclear export, though they are plausible. While the authors make clear at the end of section 10 that CpG frequencies and GC content are not the same, they do not make this distinction earlier and this could confuse the reader regarding potential binding of ICP27 and ORF57 to CpG dinucleotides.

Author Response

Major comments

  1. One way to improve this review would be to better balance the author’s paper on ZAP restriction of HCMV (Reference 34) with the paper from the Brinkmann lab (Reference 39). There are important differences in the conclusions drawn by the two groups, but sections 6 and 7 ‘Evasion of ZAP by Beta-herpesviruses’ seem to only discuss the author’s work on this topic. The different conclusions for which HCMV transcripts are targeted by ZAP need to be discussed as well as the importance of CpG dinucleotides for ZAP binding and viral RNA degradation.

Response: We agree and have included an additional section discussing ref 39 (lines 290-291).

  1. It is not currently generally accepted that ZAP only targets CpG dinucleotides in viral RNA; other nucleotide motifs may also be important. The authors reference the appropriate papers by the Simmonds lab (references 16 and 32) that show that ZAP can target UpA dinucleotides in positive strand RNA viruses, but do not appear to take this into account when they imply that CpG dinucleotides are the determinant for ZAP antiviral activity. They also do not discuss work on HIV-1 that indicates the position, local sequence context or RNA structure appears to determine that CpGs in env are targeted more efficiently by ZAP than CpGs in other regions of the viral genome (Kmiec et al 2020 mBio. 11:e02903-19 and Ficarelli et al 2020 J Virol. 94:e01337-19) or that in vitro binding assays have identified an extended motif for ZAP binding (Luo et al 2020 Cell Rep. 30:46-52.e4). These points also influence the discussion in sections 9 and 10, which need to consider that nucleotide motifs other than CpG may influence ZAP antiviral activity and not all CpGs may be equally targeted by ZAP.

Response: We agree, this was an important omission from the first draft and we thank both reviewers for pointing this out. We have now extended the section on ZAP to address this (lines 136-162)

  1. In section 4 ‘CpG profiles of herpesviruses', the authors discuss the CpG dinucleotide frequencies within the herpesvirus DNA genomes. However, for restriction by ZAP, CpG frequencies in the RNA is the relevant factor. What is known about the CpG frequencies in hepesvirus RNAs and can this section include that information?

Response: We have now included a graph of GC levels in individual human herpesvirus open reading frames and included text to explain that, as the majority of the viral genome is transcribed, genome content and transcriptome content are intimately linked. We have also been more specific when discussing nucleotide content of transcripts versus genomes.

  1. Sections 9 and 10 are speculative and there does not seem to be much direct experimental evidence that supports the ideas regarding CpG dinucleotides and HCMV mRNA splicing and nuclear export, though they are plausible. While the authors make clear at the end of section 10 that CpG frequencies and GC content are not the same, they do not make this distinction earlier and this could confuse the reader regarding potential binding of ICP27 and ORF57 to CpG dinucleotides.

Response: We do point this out in lines 320-321 and have now added text in lines 434-436 in section 9 to highlight the difference and to avoid confusion.

We have also made substantial additional edits to improve the overall structure and flow of the article.

Reviewer 2 Report

Synposis
Within the context of innate immunity, a range of host cell proteins counteract viral infections in a non-specific way. Among them is the product of the ZC3HAV1 gene, called ZAP. This protein is commonly thought to target CpG rich sequences, which are abundant in some (but by far not all) viruses. Initially, the antiviral activity of ZAP has been discovered in the context of retrovirus inhibition, however has been described to act against a range of both DNA and RNA viruses. In this review, the authors, who have recently published a research article on the role of ZAP in CMV infection, summarize current knowledge about ZAP function in repressing herpesviruses, and relate ZAP activity to the evolution of CpG content in viral genomes.
The review addresses an important and evolving topic. The manuscript is very well written, and in my opinion clearly deserves publication. However, given that the mechanisms of antiviral activity by ZAP are not well known (e.g. direct RNA interactors are just starting to be characterized), there might be more ways in how ZAP activity and CpG content of Herpesviruses are related to each other. This particularly concerns sections 5, 6 and 8, as detailed below. 

Major issues
1.    Lines 68-73: Indeed, the GC content in the human genome can be very variable, but since the topic is an RNA binding protein, it is important to distinguish between GC content of the transcriptome and that of the genome. 
2.    Along these lines, Figure 1 should be extended with a violin plot of the GC content of the transcriptomes, including the human one.
3.    Line 112, “Until recently…” – in my opinion the lack of knowledge about binding sites on target RNA is one of the biggest gaps in the current ZAP knowledge, and therefore this section could be expanded a bit (see also later), e.g. by further specifying, beyond Ref 18, how the binding sites were mapped (reporter assays in Ref. 30, CLIP experiments in Ref. 39, etc.).
4.    Line 115/129: within the text, TRIM25 comes in line 115, however only in line 129 it is mentioned that it is a co-factor. Consider shifting the part in lines 112 to 120 to after 138. 
5.    Please include the human transcriptome in Figure 2.
6.    In section 5-7, the authors rely on a hypothesis, that, although it may be true, is for the moment a bit too narrow, namely that all CpGs are targeted by ZAP, and therefore CpG reduction itself already evades ZAP. In Ref. 18 (Fig. 4a) it looks like all binding sites are next to CpG, but not all CpG are bound by ZAP (e.g. region around position 7800). In Ref. 39, the authors show that there is a ZAP specificity for UL4/UL5, although there are lots of other CpG containing transcripts. This points to a more complex binding mechanism, which could depend on secondary motifs, structural features, RNA modifications, co-factors etc., and ZAP evasion could then also involve these. The observation of CpG suppression primarily in IE1 and the explanation in lines 277 onwards is indeed very elegant, however at the moment more options should be left on the table. Noteworthy, in their own previous paper, point out that “High ZAP expression correlates with failure in HCMV acute replication.”, without showing causality.
7.    Lines 233 to 239, single-cell experiments (Refs. 50 and 51) could indeed be helpful here, however the selection of references seems a bit arbitrary given that there are several CMV single-cell experiments published or on biorxiv. Since these lines are not directly helpful for the understanding of ZAP, they could be deleted.
8.    Sections 8/9 brings up several interesting thoughts, concepts and questions. In part, it is not so well structured. I guess the basic idea here is to argue that the CpG content is related to the splicing pattern, i.e. to include efficient splicing into the trade-off considerations between favorable aspects of CpG content and detrimental ZAP activity. This is a very interesting and important thought. I would suggest to rewrite these sections with two aspects. First, it could be enlarged by encompassing other aspects of RNA metabolism next to splicing, such as translation efficiency, interactions with RNA binding proteins (mentioned for ICP27), RNA modifications, structures etc., which need to be taken into account for this tradeoff discussion (which includes a bit what is written on lines 444-449). Second, some aspects are missing, such as that there are different cellular mechanisms to export unspliced RNAs (histone hairpins etc.), or the interplay between CpG content/methylation and nucleosome/chromatin structure (e.g. https://www.ncbi.nlm.nih.gov/pmc/articles/PMC2926781/). 

Minor issues
1.    The authors should cite the review https://www.mdpi.com/1422-0067/22/14/7503/htm which just came out (so was not published at the time of writing this manuscript),  and which in section 2.3. also contains a a summary of current ZAP/herpesvirus knowledge.
2.    For clarity, the official gene name ZC3HAV1 as well as the other frequently used name PARP13 should be mentioned.
3.    Line 101, the sentence “…presumably due to IFN dependent differential splicing” could be misleading, rephrase to e.g. “differential splicing downstream of IFN signaling”.
4.    Line 110, there is now quite some interesting literature for ZAP on SARS-CoV-2, which could also be cited.
5.    Line 284, it could be noted that Herpes simplex viruses have mechanisms to even suppress ZAP transcription (https://www.nature.com/articles/s41598-020-77725-4).

Author Response

Major issues
1.    Lines 68-73: Indeed, the GC content in the human genome can be very variable, but since the topic is an RNA binding protein, it is important to distinguish between GC content of the transcriptome and that of the genome.

Response: We agree and have addressed this in figure 1 and lines 76-79.

  1.    Along these lines, Figure 1 should be extended with a violin plot of the GC content of the transcriptomes, including the human one.

Response: We have included dot plots as this gives more granularity in the data and maintains continuity with figure 2. We attempted to include the human transcriptome, but it dwarfed the data from the virus and proved impractical – perhaps we misunderstand the reviewers suggestion? We have included the average GC content as a separate bar in figure 1B.

  1.    Line 112, “Until recently…” – in my opinion the lack of knowledge about binding sites on target RNA is one of the biggest gaps in the current ZAP knowledge, and therefore this section could be expanded a bit (see also later), e.g. by further specifying, beyond Ref 18, how the binding sites were mapped (reporter assays in Ref. 30, CLIP experiments in Ref. 39, etc.).

Response: Agreed, we have now extended the section on ZAP to include this information.

  1.    Line 115/129: within the text, TRIM25 comes in line 115, however only in line 129 it is mentioned that it is a co-factor. Consider shifting the part in lines 112 to 120 to after 138.

Response: Agreed, the structure was confusing. We have now removed TRIM25 from this line and instead describe its role in full later in the section.

  1.    Please include the human transcriptome in Figure 2.

Response: We found the same problem as figure 1. Again, we may be misunderstanding the reviewers suggestion and are unsure of how to successfully format such a figure.

  1.    In section 5-7, the authors rely on a hypothesis, that, although it may be true, is for the moment a bit too narrow, namely that all CpGs are targeted by ZAP, and therefore CpG reduction itself already evades ZAP. In Ref. 18 (Fig. 4a) it looks like all binding sites are next to CpG, but not all CpG are bound by ZAP (e.g. region around position 7800). In Ref. 39, the authors show that there is a ZAP specificity for UL4/UL5, although there are lots of other CpG containing transcripts. This points to a more complex binding mechanism, which could depend on secondary motifs, structural features, RNA modifications, co-factors etc., and ZAP evasion could then also involve these. The observation of CpG suppression primarily in IE1 and the explanation in lines 277 onwards is indeed very elegant, however at the moment more options should be left on the table. Noteworthy, in their own previous paper, point out that “High ZAP expression correlates with failure in HCMV acute replication.”, without showing causality.

Response: We agree with this point. We have extended the section on ZAP targeting and moved sections on alternatives to the role of ZAP in shaping CpG content from the conclusion section on methylation and codon bias to section 8 and have included a new section on RNA editing. We have also addressed the CLIPSeq data in ref 39. We have acknowledged the necessity for additional studies to confirm the role of CpG suppression in IE1.

  1.    Lines 233 to 239, single-cell experiments (Refs. 50 and 51) could indeed be helpful here, however the selection of references seems a bit arbitrary given that there are several CMV single-cell experiments published or on biorxiv. Since these lines are not directly helpful for the understanding of ZAP, they could be deleted.

Response: Agreed, these lines have now been deleted.

  1.    Sections 8/9 brings up several interesting thoughts, concepts and questions. In part, it is not so well structured. I guess the basic idea here is to argue that the CpG content is related to the splicing pattern, i.e. to include efficient splicing into the trade-off considerations between favorable aspects of CpG content and detrimental ZAP activity. This is a very interesting and important thought. I would suggest to rewrite these sections with two aspects. First, it could be enlarged by encompassing other aspects of RNA metabolism next to splicing, such as translation efficiency, interactions with RNA binding proteins (mentioned for ICP27), RNA modifications, structures etc., which need to be taken into account for this tradeoff discussion (which includes a bit what is written on lines 444-449). Second, some aspects are missing, such as that there are different cellular mechanisms to export unspliced RNAs (histone hairpins etc.), or the interplay between CpG content/methylation and nucleosome/chromatin structure (e.g. https://www.ncbi.nlm.nih.gov/pmc/articles/PMC2926781/).

Response: Agreed, we have now extended this section to include alternative explanations for CpG dinucleotide patterns and have expanded the section on RNA export to included different pathways used for exporting unspliced RNAs.

We have also made additional extensive edits to improve the structure and flow of the article.

Minor issues
1.    The authors should cite the review https://www.mdpi.com/1422-0067/22/14/7503/htm which just came out (so was not published at the time of writing this manuscript),  and which in section 2.3. also contains a a summary of current ZAP/herpesvirus knowledge.
2.    For clarity, the official gene name ZC3HAV1 as well as the other frequently used name PARP13 should be mentioned.
3.    Line 101, the sentence “…presumably due to IFN dependent differential splicing” could be misleading, rephrase to e.g. “differential splicing downstream of IFN signaling”.
4.    Line 110, there is now quite some interesting literature for ZAP on SARS-CoV-2, which could also be cited.
5.    Line 284, it could be noted that Herpes simplex viruses have mechanisms to even suppress ZAP transcription (https://www.nature.com/articles/s41598-020-77725-4
).

Response: We have addressed each of these issues, except for point 5. We felt the context of this paper was too broad to be specifically included for inhibition of ZAP activity.

Round 2

Reviewer 1 Report

The revised manuscript is an interesting review of how ZAP regulates herpesvirus replication and genome characteristics.

Author Response

We thank the reviewer for their helpful comments.

Reviewer 2 Report

In my opinion, the authors have addressed the concerns of both reviewers quite well. Only two minor issues remain to be addressed in a final revision.

Remaining issues
1.    The authors have indeed, as also requested by the other reviewer, better balanced their own paper on ZAP restriction of HCMV with the paper from the Brinkmann lab. However, the abstract should also be changed correspondingly. A suggestion would be to write “We and others” in line 23, and then omitting the second part of the sentence “which in turn evades ZAP…”, and also the following sentence “While suppression…unspliced transcripts”.

2.    The figures indeed look much nicer, and of course a beeswarm/dot plot with the human data would look bad. This is why I suggested a violin plot, which is independent on the number of datapoints. The authors could for the viruses underlay the current dot plot with violin plots, and for the human data only show the violin. The reason I asked for the plot is actually because, as far as I remember, the suppression of CpG is quite strong in the human transcriptome, which would support the basic assumption that in some ways high CpG is beneficial for the viruses.

Author Response

  1. We have changed the text in the abstract to a neutral statement, which also enables the subsequent statements to remain the same.
  2. We thank the reviewer for the suggestion and we are now able to supply the figures with both bee swarm plots for the viral dinucleotide frequencies with violin plots for GC content and dinucleotide frequency for the  human Transcriptome. We agree this adds value to the figures.